# Getting Recovery Right After Neck Dissection (GRRAND-F): mixed-methods feasibility study to design a pragmatic randomised controlled trial protocol

Victoria Gallyer ,[1] Toby O Smith ,[1,2] Beth Fordham,[1] Susan Dutton,[3] Mae Chester-Jones,[3] Sarah E Lamb,[1,4] Stuart C Winter,[5] On behalf of the GRRAND-F Trial Collaborators

► Prepublication history and additional online supplemental material for this paper are available online. To view these files, please visit the journal online (http://dx.doi.org/10.1136/bmjopen-2020-045741).

For numbered affiliations see end of article.

Correspondence to
Stuart C Winter;
stuart.winter@nds.ox.ac.uk

## ABSTRACT

**Introduction** We will evaluate the feasibility of a randomised controlled trial to estimate the effectiveness and cost-effectiveness of a rehabilitation intervention on pain, function and health-related quality of life following neck dissection (ND) after head and neck cancer (HNC).

**Methods and analysis** This is a pragmatic, multicentred, feasibility study. Participants are randomised to usual care (control) or usual care plus an individualised, rehabilitation programme (Getting Recovery Right After Neck Dissection, GRRAND intervention). Adults aged over 18 with HNC for whom ND is part of their care will be recruited from specialist clinics. Participants are randomised in 1:1 ratio using a web-based service. The target sample size is 60 participants. Usual care will be received by all participants during their postoperative inpatient stay consisting standard National Health Service care supplemented with a booklet advising on postoperative self-management strategies. The GRRAND intervention programme consists of usual care plus up to six individual physiotherapy sessions including neck and shoulder range of motion (ROM) and progressive resistance exercises, advice and education. Between sessions participants will be advised to complete a home exercise programme. The primary outcome is to determine recruitment and retention rates from study participants across sites. Outcomes will be measured at 6 and 12 months. Participants and physiotherapists will be invited to an optional qualitative interview at the completion of their involvement in the study. The target qualitative sample size is 15 participants and 12 physiotherapists. Interviews aim to further investigate the feasibility and acceptability of the intervention and to determine wider experiences of the study design and intervention from patient and physiotherapist perspectives.

**Ethics and dissemination** Ethical approval was given on 29 October 2019 (National Research Ethics Committee Number: 19/SC/0457). Results will be reported at conferences and in peer-reviewed publications.

**Trial registration number** ISRCTN11979997.

**Status** Trial recruitment is ongoing and is expected to be completed by 30 August 2021.

## Strengths and limitations of this study

► Getting Recovery Right After Neck Dissection is a pragmatic, multicentred, randomised control feasibility trial.
► We will evaluate whether it is feasible to run a randomised controlled trial to assess the effectiveness and cost-effectiveness of a rehabilitation intervention in improving pain, function and health-related quality of life following neck dissection after head and neck cancer.
► The primary outcome is recruitment and retention rates.
► The qualitative substudy will explore the wider experiences and perceptions of the study design and intervention from a patient and physiotherapist perspective.

## INTRODUCTION

Head and neck cancer (HNC) affects 700 000 people worldwide and over 11 000 in the UK annually.[1–3] HNC refers to neoplasms at different anatomical sites. Within the UK, tumours of the oropharynx are the most common and have seen a twofold increase in incidence over the last 20 years, largely attributed to human papillomavirus.[4 5] During this time, there has also been a 30% increase in oral cancer.[4–6] While there has been a significant increase in HNC, prognosis and survival in the UK continues to improve.[4 6] Therefore, the proportion of people living with the effects of this cancer and its treatment continues to increase.

The treatment pathway for HNC is complex, due to the varied anatomical sites of disease and the needs of the patient. Treatment for HNC requires treatment of the primary site, as well as the neck when there is spread to the

BMJ

lymph nodes or high probability of spread. Historically almost all patients received a neck dissection (ND). With the advent of chemoradiotherapy as a curative treatment, less patients require an ND. However, even with this approach, up to 20% of patients require an ND due to residual disease.[6] Side effects from surgery can be significant, including swallowing problems, neck and shoulder problems, difficulties sleeping, fatigue and anxiety.[7 8]

Postoperative complications are common following ND.[8–11] Early complications can include shoulder pain and infection. Late complications may not appear until 3 months post-treatment, and can continue to present over 5 years.[12 13] These complications include shoulder movement dysfunction, speech, swallowing and musculoskeletal problems such as cervical contracture and muscle wastage.[12] Psychosocial complications are also highly prevalent postoperatively, predominantly fatigue, anxiety, depression, sleep disturbance and social isolation. Sequelae of shoulder dysfunction and psychosocial complications are strongly associated with reduced return to work, with up to 50% of patients ceasing working due to shoulder disability alone.[10 14]

Rehabilitation was one of 22 key questions in the 2016 National Institute for Health and Care Excellence (NICE) Clinical Guideline[15] on the management of HNC. The guideline recommends clinicians 'consider progressive resistance training for people with impaired shoulder function, as soon as possible after ND'. The review noted that this evidence was from small trials with a high risk of bias. The review also highlighted a knowledge gap on how to rehabilitate HNC patients' wider side effects. The NICE guideline concluded that a prospective randomised trial was required to understand how best to promote recovery following HNC, making this a recognised National Health Service (NHS) research priority.[15]

Currently, there is no national standard best practice for rehabilitation following HNC. Our study development work[16] and feedback from patient and public involvement (PPI) representatives has shown that physiotherapy practice varies across the UK. The findings suggested that rehabilitation in the form of physiotherapy is not routinely available to patients with HNC, in either inpatient or outpatient settings.[16] When rehabilitation is offered it is often not evidence based, and targets acute respiratory care, ROM exercises for the neck and shoulder, and advice on positioning of the upper limb and shoulder girdle.[15] A booklet may be provided to supplement this treatment. Outpatient treatment is minimal, and most commonly reactive, driven by patient request. While trials have begun to provide indicative findings on different rehabilitation strategies for this population,[17 18] the current evidence base is limited in quality and only focuses on shoulder exercises. There remains a gap in knowledge on how to rehabilitate patient's wider side effects following surgery for HNC such as fatigue, anxiety, poor sleep and return to work. Consequently, both Cochrane[19] and NICE[15] concluded that further high-quality research is needed to determine how best to promote recovery for shoulder function, quality of life and cost-effectiveness of treatment.

This study will evaluate whether it is feasible to conduct a randomised controlled trial (RCT) to assess the effectiveness and cost-effectiveness of a multimodal rehabilitation intervention in improving pain, function and health-related quality of life following ND after HNC. In addition to investigating the feasibility of an enhanced rehabilitation intervention following HNC ND, this trial will also standardise usual care.

## METHODS AND ANALYSIS

### Trial design
A mixed-methods feasibility study investigating the design of an RCT to test the clinical and cost-effectiveness of usual care and an individualised, rehabilitation programme (GRRAND) compared with usual care alone in patients undergoing an ND for HNC. The study flow chart is presented as figure 1. Table 1 presents a summary of trial objectives, outcome measures and time points.

### Eligibility
Participants are eligible to take part in the trial if they fulfil the eligibility criteria listed in box 1. All patients having an ND regardless of other associated procedures are eligible. HNC can arise at a number of anatomical sites and an ND is often combined with additional treatment such as radiotherapy to the primary site. This reflects the expected practice in HNC treatment.[15] We will record the location of cancer, specific surgical interventions and planned additional treatments such as radiotherapy, to ascertain the profile of the recruited ND cohort. This will provide information to aid sample size calculations, stratification approaches and analysis plans for confounders/modifiers in a definitive trial.

### Recruitment
Potential participants will be identified from UK NHS hospital trusts as requiring an ND as part of their treatment, and will be approached by a member of the clinical team to ask whether they would like to know more about the GRRAND-F study.

They will be asked to read the patient information sheet (PIS) and to discuss their potential participation with anyone who they feel would provide useful advice. Potential participants will also be provided with contact information for the research team who will be able to answer any questions relating to the study. The number of patients provided with the PIS will be recorded to monitor the number of patients who are approached.

Eligible patients who agree to participate will then be asked to provide their written informed consent (online supplemental file 1).

### Randomisation, blinding and allocation concealment
Following the completion of the consent process baseline data will be collected. Participants will then be randomised

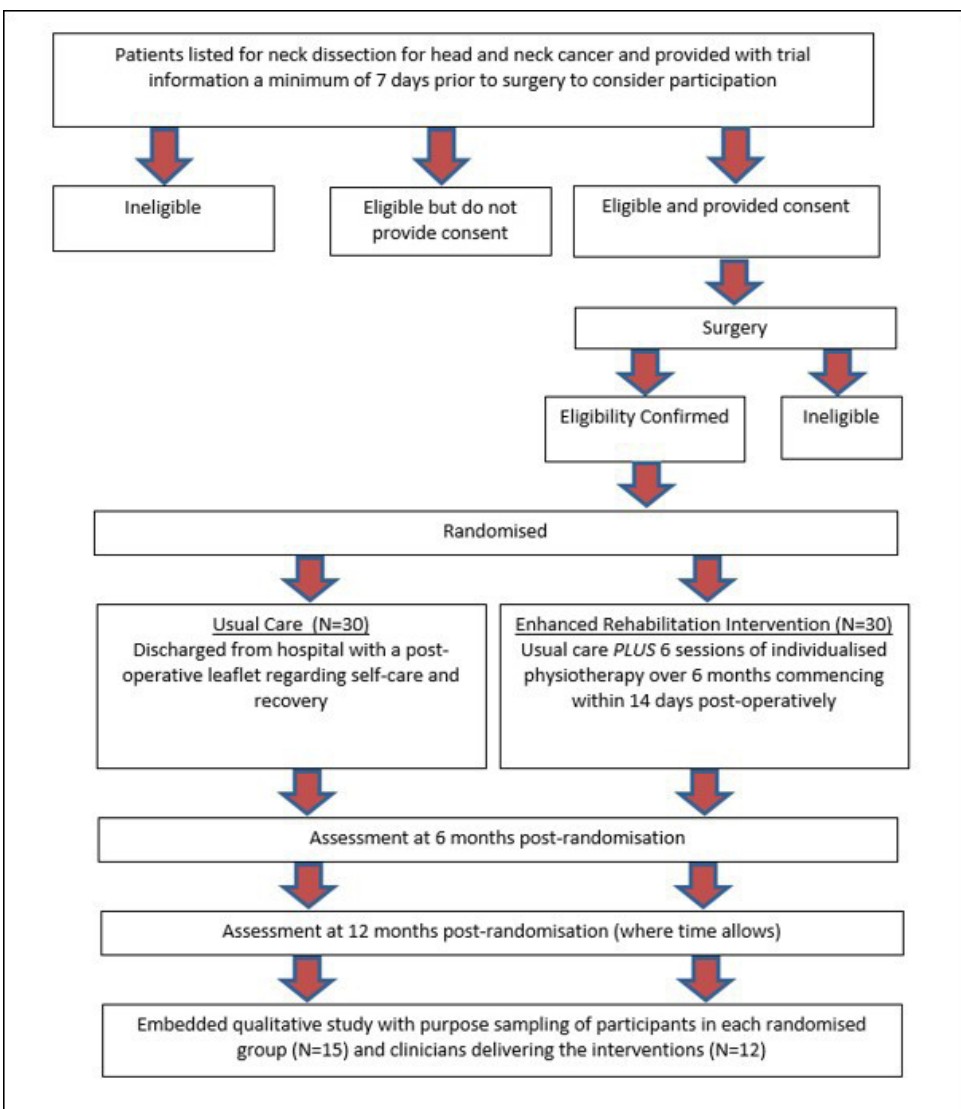

**Figure 1** Study flow chart.

once their eligibility has been confirmed post-operatively prior to hospital discharge.

Participants will be randomised in a 1:1 ratio using the centralised web-based randomisation service provided by Oxford Clinical Trials Research Unit. Randomisation will be undertaken using minimisation to ensure balanced allocation of participants across the two treatment groups, stratified by hospital site and spinal accessory nerve sacrifice.

The minimisation algorithm will incorporate a non-deterministic element and will be seeded using simple randomisation to prevent predictability in the early stages of the study.

Due to the nature of the intervention, participants and clinicians delivering physiotherapy will not be blinded to treatment allocation.

### Intervention
#### Usual care
Usual care will be received by both control and experimental intervention groups.

As part of usual care, all participants will receive the same in-patient rehabilitation programme, commencing day one postoperatively (or next physiotherapy working day), consisting of:
1. Advice to practise simple ROM exercises for the face and neck for the purpose of preventing the onset of postsurgical contracture and optimising swallowing and shoulder movement.
2. Respiratory care, targeting sputum clearance and breathing control.
3. Education on body positioning to reduce pressure and pull on the shoulder girdle, oral health to reduce food pocketing in the mouth, and pain management and pacing activities to optimise levels of comfort and function.

The content, dosage and timing of in-patient physiotherapy contact will be recorded.

Reflecting usual care, on discharge participants will receive a booklet providing advice on postoperative

**Table 1** GRRAND-F objectives, outcome measures and measurement time points

| Objectives | Outcome measures | Time points |
|---|---|---|
| **Primary objective** | | |
| To determine recruitment and retention rates from study participants across sites. | Study recruitment screening logs, consent forms and logs of data collection forms completed at each time point. | 6 months and 12 months (for those participants who reach this time point within the study window). |
| **Secondary objectives** | | |
| To determine potential risks of intervention contamination. | Intervention logs and qualitative interviews (face to face with patients/telephone based with physiotherapists). | Completion of intervention and qualitative interviews. |
| To determine feasibility and acceptability of the intervention from patient and physiotherapist perspectives. | Intervention log, cross-over event as reported in protocol deviation forms, attrition rate and 'did not attend' rates for intervention. Qualitative interviews. Safety reporting forms. | Completion of intervention and qualitative interviews. |
| To estimate the sample size calculation for a definitive trial. | Expected primary and secondary outcome measure: Shoulder Pain and Disability Index (overall and pain and function sub-scales); EQ-5D-5L; EORTC quality of life questionnaire (C30 core and disease-specific H&N43); health resource use questionnaire; adverse events; shoulder/neck range of motion and grip strength. | At the end of the trial. |
| To determine wider experiences and perceptions of the study design from a patient and physiotherapist perspective. | Qualitative interviews. | Completion of the qualitative interviews. |

EORTC, European Organisation for Research and Treatment of Cancer; GRRAND-F, Getting Recovery Right After Neck Dissection; H&N43, Head and Neck 43 Questionnaire.

self-management strategies including exercise, pain management, return to work and activities of daily living. This has been developed by the multidisciplinary trial team and collaborations with two of the participating NHS centres in Birmingham and Oxford to ensure that the information is standardised. Reflecting current practice, once discharged from hospital, physiotherapy will not be routinely provided to these participants.

### Experimental intervention

Participants randomised to this group will receive the same in-patient rehabilitation programme as participants in the Usual Care Group PLUS an individualised

---

**Box 1  Getting Recovery Right After Neck Dissection (GRRAND-F) eligibility criteria**

**Inclusion criteria**
- ► Aged 18 years and above.
- ► Being treated for head and neck cancer in whom a neck dissection is part of their care.
- ► Willing and able to provide informed consent.
- ► Able to understand written english.
- ► Participant is willing to attend the physiotherapy outpatient department if randomised to the experimental intervention arm (GRRAND-F intervention).
- ► Who remain eligible postoperatively when reviewed prior to randomisation.

**Exclusion Criteria**
- ► If treatment is palliative (expected survival 6 months or less).
- ► Those with a pre-existing, long-term neurological disease affecting the shoulder, for example, hemiplegia.
- ► Cognitive impairment (defined as an Abbreviated Mental test score of 7 or less).

---

rehabilitation programme. This will be delivered by a GRRAND-F-trained physiotherapist in an outpatient setting. In the event that the participant is still an inpatient, this will be commenced in hospital and continued, postdischarge, in an outpatient setting. The frequency to which this change of setting occurs will be recorded as part of the feasibility outcomes.

At the initial consultation, physiotherapists will assess the participant to identify modifiable physical and psychosocial factors associated with poor recovery following HNC surgery. These may include: muscle weakness, limited ROM, reduced sensation, pain and fear avoidance beliefs. Based on this assessment, physiotherapists will prescribe from a prespecified range of rehabilitation options (see figure 2).

Programmes will be individualised to contain one, several or all of the treatment options, dependent on participant's needs. Participants will also be provided with a home exercise programme to supplement face-to-face sessions.

### Individualised rehabilitation options

1. ROM exercises targeting muscles and joints of the face, neck and shoulder impacted by ND. The purpose of these exercises is the prevention of postsurgical contracture, and the maintenance of swallowing and upper limb mobility.
2. Progressive resistance exercises, targeting strengthening of the neck and shoulder. Resistance loads will initially be set at a moderate level of exertion (based on the modified Borg scale of perceived exertion[20]) to permit progression, enhance motivation and adherence, and reduce the possibility of symptom flare-up. Resistance will consist resistance bands at the shoulder

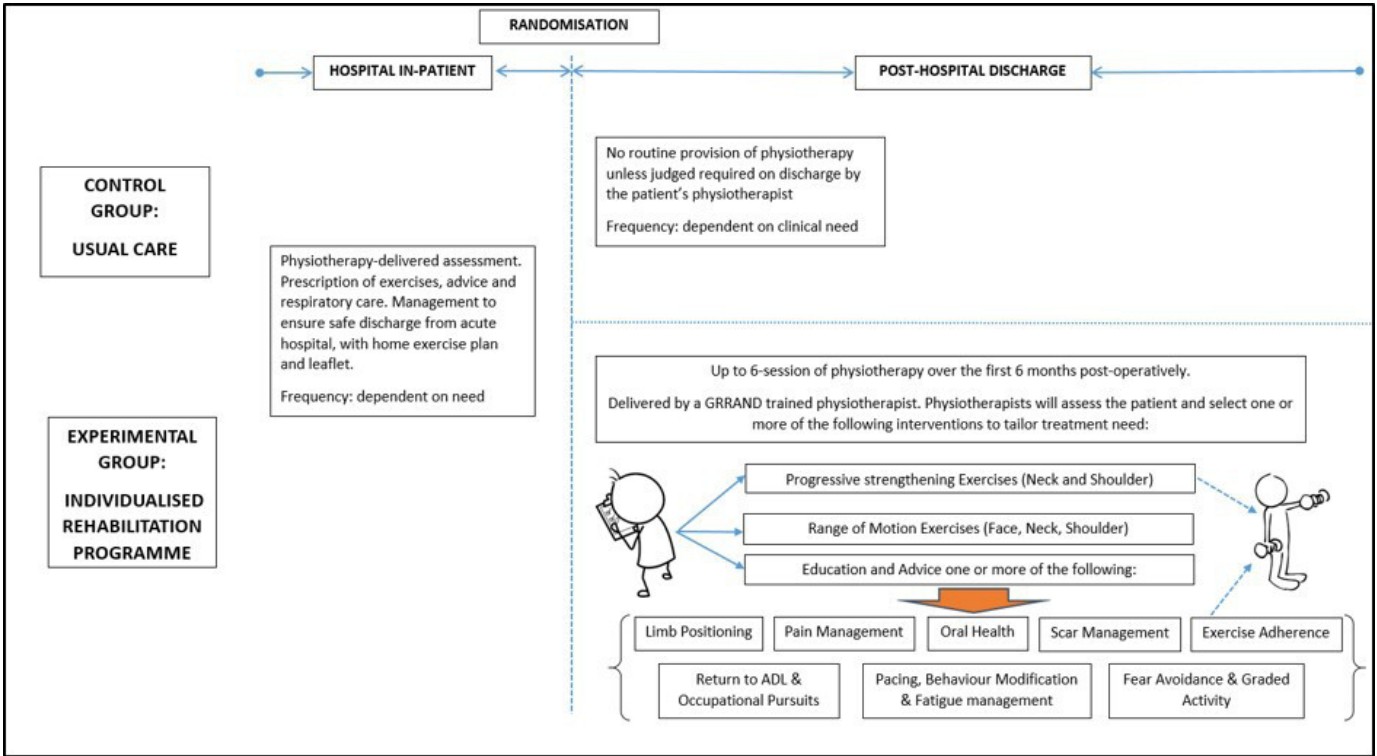

**Figure 2** GRRAND-F intervention schema. ADL, activities of daily living; GRRAND-F, Getting Recovery Right After Neck Dissection.

and isometric resistance provided by the participant's hand for neck and temporomandibular joint exercises.

Exercises will be progressed by increasing the resistance load, speed, number of repetitions and sets or by progressing the range in which the exercise is completed and through the introduction of weight-bearing exercises through the upper limb. Additionally, the exercises will become increasingly 'task specific', targeting participant's specific functional goals.

3. Education and advice on a number of recognised potential postoperative complications including:
   – Positioning limbs to prevent joint contractures.
   – Oral health particularly for patients following upper cervical/head/oral surgery.
   – Pain management for both early and later postoperative stages through positioning, taking prescribed analgesics and pacing/behaviour modfication.
   – Scar management.
   – Exercise adherence and return to function with fatigue management and pacing of activities.
   – Promote independence and confidence to return to normal activities of daily living, work and social pursuits.

This will be delivered through the introduction of techniques of goal setting, fear avoidance, pacing and fatigue management, behaviour modification and graded activity. This has been successfully taught and delivered by the research team in previous National Institute for Health Research trials (BOOST,[21] DAPA[22]), to provide a basis for this new intervention. Advice will be provided through discussion during consultations and re-enforced with worksheets designed by the multidisciplinary trial team.

The intervention may be modified in the development phase of the trial. The intervention will be finalised prior to the main trial. If there are no substantive changes, participants will contribute to the main trial analysis.

### Delivery

The experimental intervention will be delivered a maximum of six sessions over a 6-month period. The design will enable assessment of how many sessions are required. The first session will aim to occur within 14 days of surgery. Reflecting normal NHS practice, the initial session will be 60 min, and subsequent sessions up to 45 min in duration. The physiotherapist, in collaboration with the participant, will agree the spacing of sessions, reflecting normal clinical practice. This spacing will allow for maximum progression of the intensity of exercise over a time period sufficient to (hypothetically) produce an improvement in outcome. Treatment options may also be added or removed at each session, in line with the participant's current treatment progress and health status.

The timing and spacing of sessions around additional treatments such as radiotherapy and chemotherapy will be determined by the participant and physiotherapist. Through this, if the participant or physiotherapist feel that the intervention is not appropriate due to radiotherapy/chemotherapy side effects such as fatigue, pain or nausea, the GRRAND intervention will be delayed until symptoms reduce. Alternatively, if the participant and physiotherapist agree that the GRRAND intervention

would be beneficial alongside such treatments, this will be permitted. This reflects the individualised nature of the intervention.

## Contamination

The GRRAND-F physiotherapists who deliver the experimental intervention sessions where possible will not deliver physiotherapy to those in the control group (and vice versa). The details of the physiotherapists delivering sessions will be recorded and reviewed to monitor this risk of contamination. Due to the interventions being individualised and delivered in an outpatient setting, there is a low risk of participants sharing their knowledge and experience between groups, further minimising the risk of between-group contamination.

## Cointerventions

Respecting the pragmatic nature of this study design, participants from either group will not be asked to desist from receiving any other forms of treatment during the trial or follow-up periods. If a participant receives additional treatment, the details of the treatment received and the reasons for administering will be collected.

## Quality assessment

The trial will be monitored and audited in accordance with the current approved protocol, good clinical practice,[23] relevant regulations and standard operating procedures.

All designated physiotherapists who deliver usual care will be taught the standardised control intervention procedures.

Physiotherapists delivering the GRRAND intervention will attend a face-to-face training session where they will be taught the intervention and processes involved by a member of the GRRAND-F team who developed the intervention (TOS and VG). Each intervention physiotherapist will be monitored during a site visit at their third/fourth session. Sessions will be monitored against the protocol to determine whether there are issues around fidelity, contamination across groups or adherence/compliance of participants. Where further training or further monitoring visits are required, these will be instigated following these visits.

## Assessments

Data will be clinical and participant-reported and collected using questionnaires at baseline and 6 months postrandomisation. Data will also be collected for those participants who reach 12-month follow-up during the data collection phase. This is estimated to be applicable for up to 50% of the cohort. Data will be collected alongside routine clinical appointments at each site. A primary endpoint of 6 months postrandomisation was chosen to provide a signal on clinical outcomes after completing the intervention. The 12-month data provide data to assess the risk of attrition and missing data at 12 months, which will assist with the development of the definitive trial if it proves to be feasible.

## Baseline assessment

Baseline data will be collected prior to randomisation once consent has been obtained, typically during the preoperative assessment. Data collection is described in table 2.

Outcome data to be collected at each of the data collection intervals are listed below.
- ► Shoulder pain and function measured using the well-validated Shoulder Pain and Disability Index (SPADI).[24 25]
- ► Pain measured using the SPADI 5-item Pain Subscale[25] and a Numerical Rating Scale.
- ► Function measured using the SPADI 8-item Function subscale[25]
- ► Pain medication details and usage relating to head, neck and shoulder.
- ► Chemotherapy and radiotherapy treatment provision.
- ► Health-related quality of life measured using the EQ-5D-5L score[26] and the EORTC questionnaires (C30 (core)[27] and H&N43 (head and neck specific)[28 29]).
- ► Health resource use questionnaire (collection of health resources for computation of direct medical, direct nonmedical and indirect costs); additional out-of-pocket expenses and work absence.
- ► Physical performance measures including goniometer-measured shoulder and neck active ROM and hand-held dynamometer-measured grip strength will be measured by an appropriately trained member of the research team.
- ► Adverse events: such as prolonged delayed onset muscle soreness, swelling and wound irritation.

## Follow-Up procedures

Data will be collected from participants at 6 and 12 months (if applicable) from date of surgery with a target of ±1 month, at their routine NHS check-up appointments. If participants do not attend their follow-up appointment, they will be contacted by telephone, and, if appropriate, sent the questionnaires to complete. The study team will attempt to telephone these participants on up to two occasions. If these methods fail, we will categorise the participant as a 'non-responder' for that time-point only. The data collection schedule is presented in table 2.

## Outcome measures

Feasibility outcome data to be collected will include:
- ► Screening log numbers of eligible patients, including reasons for exclusion/non-participation.
- ► Recruitment numbers and rate; overall and per site.
- ► Protocol adherence, including fidelity to control and experimental interventions using treatment logs, timing and location of intervention delivery (in particular the first session) alongside frequency of physiotherapy contact. This will assist in assessing both potential between-group contamination and intervention delivery. We will also monitor the intervention delivery as part of the Quality Assurance monitoring visits. The findings of these visits will provide data on

**Table 2** Data collection schedule

| Data | Baseline | Inpatient predischarge | Intervention period | 6* months postrandomisation | 12* months postrandomisation |
|---|---|---|---|---|---|
| Age (years) | √ | | | | |
| Gender | √ | | | | |
| Weight (kg)/(stone/lbs) | √ | | | | |
| Height (cm)/(ft/inches) | √ | | | | |
| Ethnicity | √ | | | | |
| Drinking status | √ | | | | |
| Smoking status | √ | | | | |
| Primary cancer site | | √ | | | |
| Stage of tumour | | √ | | | |
| Neck nodal status | | √ | | | |
| Pre-existing shoulder or neck musculoskeletal disorder | √ | | | | |
| Hand dominance | √ | | | | |
| AMTS | √ | | | | |
| List of medical co-morbidities | √ | | | | |
| Employment status and current occupation (when appropriate) | √ | | | √ | √ |
| Shoulder Pain and Disability Index | √ | | | √ | √ |
| Numerical rating scale pain | √ | | | √ | √ |
| EQ-5D-5L | √ | | | √ | √ |
| EORTC QLQ-C30 | √ | | | √ | √ |
| EORTC QLQ-H&HN43 | √ | | | √ | √ |
| Physical performance measures | √ | | | √ | √ |
| Pain relief medication list | √ | | | √ | √ |
| Complications, AE, SAE details of accident and emergency attendances and hospital admissions | | √ | √ | √ | √ |
| Operation date | | √ | | | |
| Operative procedure (level of ND) | | √ | | | |
| Location of HNC | | √ | | | |
| Accessory nerve sacrificed | | √ | | | |
| ASA grade | | √ | | | |
| Pathology results | | √ | | | |
| Preoperative cancer head and neck treatment | √ | | | | |
| Chemotherapy and radiotherapy treatment provision | √ | | | √ | √ |
| Intervention fidelity and cross-over logs | | | √ | | |
| Physiotherapy intervention log (physiotherapist completed) | | √ | √ | | |
| Home exercise diary (participant completed) | | | √ | | |
| Health economic/health utilisation questionnaire | | | | √ | √ |

*Each follow-up interval±1 month.
EORTC, European Organisation for Research and Treatment of Cancer; HNC, head and neck cancer; ND, neck dissection.

intervention location, fidelity to the protocol, and barriers or facilitators to provision across the sites.
► Follow-up completion rate and overall study retention in each study arm for the outcome measures highlighted above.

The primary and secondary outcome measures for this trial are presented in table 1.

## Data analysis

### Sample size

As this is a feasibility study which is not aimed to assess treatment effects, we have not undertaken a formal power sample size calculation.

Sixty participants will be recruited, based on Teare *et al*'s recommendation[30] that between 50 and 70 are required when continuous scale data outcomes are to be collected. This assumes a 10% drop-out. This will also provide sufficient data to answer our feasibility objectives with 30 participants from each group recruited. Based on 2017 data from two of the participating sites, approximately 160 potentially eligible participants were identified. Based on a conservative judgement of 45% recruitment rate for this rehabilitation trial with this cohort,[19 31 32] over 60 participants could be recruited within a 12-month period. This is within the required number to conduct this study.

### Statistical analysis

Recruitment and follow-up rates are the main drivers for the feasibility design on the basis that unless reasonable rates can be achieved no formal trial will be possible. Recruitment rate will be calculated as the number of participants randomised as a proportion of eligible participants. Rates will be estimated based on data collected and a 95% CI determined for these measures. The rate of incomplete information either due to drop-out to the interventions or non-completion of the outcome measures will be based on the number of participants randomised. The statistical analysis will also estimate, with 95% CIs, the parameters required for a formal power calculation, particularly the SD of potential outcome measures.

If the estimated recruitment and follow-up rates are such that a multicentre definitive trial is possible no formal analysis will be undertaken and data from the feasibility will be locked and carried over into the definitive trial, where funding for the definitive trial has been obtained. In this case no formal analysis of treatment efficacy will be undertaken. The definitive trial will be planned based on the data collected during this feasibility study. The mean difference, SD and effect size with between-group inferential statistical analyses will be estimated to determine direction and magnitude of effect and to inform a power calculation for a definitive trial.

The 'traffic light' system will be used as a guide for progression to a definitive trial (table 3).[33] If any of the criteria are not met, these will be discussed by the Trial Steering Committee (TSC) to decide if a definitive trial is feasible.

Descriptive statistics will be used to describe the demographics between the two groups. Clinical outcome data will be reported depending on the type of variable: for continuous variables the means and SD in each group (or median and IQR if non-normally distributed) together with the unadjusted and adjusted difference in means and corresponding 95% CIs with analysis of covariance, adjusting for baseline values (where appropriate) and stratification factors; for categorical variables, the number and percentage of participants in each category will be reported and unadjusted and adjusted ORs (for binary outcomes) together with their 95% CIs will be reported.

All results will be based on the intention-to-treat population. Protocol deviations will be reported as these are an important part of the feasibility assessment when planning the definitive trial.

| Table 3 | Progression criteria for the GRRAND-F trial | | |
|---|---|---|---|
| | **Green (Go)** | **Amber (Amend)** | **Red (Stop)** |
| Recruitment | 60 participants recruited within 12 months | 40–59 participants recruited within 12 months | <40 participants recruited within 12 months |
| Consent | ≥40% of potentially eligible participants | 20%–39% of potentially eligible participants | <20% of potentially eligible participants |
| GRRAND-F intervention fidelity | >70% participants received protocol-compliant GRRAND-F intervention | 50%–70% received intervention as randomised | <50% received intervention as randomised |
| Contamination | <5% participants in control group received GRRAND-F intervention | 5%–10% participants in control group received GRRAND-F intervention | >10% participants in control group received GRRAND-F intervention |
| Data Completion | <15% missing data at 6 months follow-up | 15%–30% missing data | >30% missing data |
| Retention | <20% attrition at 6 months follow-up | 20%–50% attrition at 6 months follow-up | >50% attrition at 6 months follow-up |

GRRAND-F, Getting Recovery Right After Neck Dissection.

## Health economics

Data on healthcare utilisation will be collected but not analysed. To answer the feasibility questions related to the health economic perspectives, we will test the completion of the health resource use questionnaire and will present the data descriptively.

## Data management

All data will be processed according to the Data Protection Act 2018[23 34 35] and all documents will be stored safely in confidential conditions. Trial-specific documents, except for the signed consent form and contact details, will refer to the participant with a unique study participant number and initials only. Participant identifiable data will be stored separately from trial data.

## Qualitative investigation

The embedded qualitative study will assess the feasibility and acceptability of the experimental and control interventions from the perspectives of those delivering (physiotherapists) and receiving (participants) the interventions. The format and delivery of the qualitative interviews are based on parameters successfully implemented in previous trials conducted by the research team (Back Skills Training Trial (BeST),[36] Better Outcomes for Older people with Spinal Trouble Trial (BOOST),[21] The Prevention of Shoulder Problems Trial (PROSPER),[37] Strengthening and Stretching for Rheumatoid Arthritis of the Hand Trial (SARAH)[38]), and UK trials involving cancer patients.[39] Specifically, participant opinion and experience of study recruitment, intervention content, timing and accessibility and barriers and facilitators to adherence will be sought. Qualitative themes identified will be used to modify the content and delivery of a future definitive trial.

## Recruitment

Fifteen participant interviews will be conducted, involving 10 participants from the experimental intervention group and five from the control group. Based on our previous trial work,[36 38] this sample size is expected to ensure data saturation across both groups, allowing for the expected larger dataset from the experimental intervention group.

All participants will be given a brief explanation of the interviews during the initial consent process. Those willing to be interviewed will indicate permission to be contacted by the qualitative researcher on the Consent Form (online supplemental file 1). It will be clarified that not all willing participants may be required for the interview study.

Participants who have agreed to be contacted for the interview will be purposively sampled by the qualitative researcher to ensure the 15 interview participants are demographically representative of the full study sample. Targeted demographics include age, ethnicity, employment status and extent of ND. We estimate that the sample will include more males than females because approximately 70% of HNC cases in the UK in males.[40]

We aim to invite two males for every one female we interview. However, if we are restricted in the number of participants available for interview, we will interview as many as available. We will highlight the sex of participants as part of our interpretation of our qualitative analysis.

The qualitative researcher will telephone the sampled participants, and answer any questions they may have about taking part in the interviews. If the participant agrees to take part, a time and date convenient to the participant will be arranged for an interview. Interviews will be conducted face to face, and occur at a location convenient to the participant, most likely in their own home.

A minimum of one physiotherapist who delivered the experimental intervention and one physiotherapist who delivered the control intervention will be interviewed from each site, until data saturation is reached. This is anticipated to occur within a maximum of 12 interviews. Each physiotherapist will be asked to read the clinician qualitative study PIS, and then to complete a consent form (online supplemental file 2). Physiotherapists who consent to participate will be contacted to arrange a suitable time to conduct a telephone interview.

## Data collection

Interviews will be conducted 4–6 weeks after a participant's final physiotherapy session. This cross-sectional time point allows exploration of the participant's study experience and adherence to home exercise in a reasonable recall period. Participant interviews will take up to 90 min. The physiotherapist interviews will take 15–30 min and will be completed within 4 weeks of intervention completion.

We conducted a brief literature review of evidence into the biopsychosocial barriers and facilitators for this patient group to return to their daily activities with acceptable quality of life. In parallel, we attended HNC patient rehabilitation groups to deepen our understanding of the patient perspective. The themes identified from the literature review and patient groups informed the semi-structured interview guide and framework. The qualitative researcher presented these to our PPI representatives and clinical experts and refined accordingly. The refined interview guide is provided in online supplemental file 3. The interview schedule will be structured in alignment with the guidance for the qualitative exploration of intervention acceptability recently published in the BMJ.[41] Interviewees will have the opportunity to suggest and/or discuss additional questions. Interviews will be audio recorded, and independently transcribed.

## Data analysis

Transcriptions will be managed using NVIVO software.[42] Qualitative researcher (BF) will analyse the data using framework analysis.[43] The analytical framework will be informed by our evidence synthesis of the biopsychosocial rehabilitation and behaviour change literature and refined through consultation with PPI and clinical

experts. After the coding of each transcript the working framework will be discussed with patient, clinical and research team members to reduce researcher bias and strengthen the framework's reliability. The final framework will include data from participants and physiotherapists and will be triangulated with quantitative data. We will produce and publish a framework of understanding for the intervention and trial progression.

## Trial status

The trial is funded for 24 months commencing in September 2019. Recruitment is expected to be complete by October 2020 with the final follow-up visit completed by April 2021. The trial will be completed by 31 August 2021. Due to the COVID-19 outbreak in the UK from March 2020, the trial timelines are expected to be extended.

## Protocol changes resulting from COVID-19

The protocol was amended to reflect the NHS service delivery changes secondary to COVID-19. These amendments include allowing intervention delivery to have the option of video consultations in line with local NHS Trusts' policies. The change to online consultations has been reflected in the addition of eligibility criterion 'When the hospital is only providing video consultation physiotherapy sessions, does the patient have access to the internet through a computer or tablet'. Video-delivered interventions will be monitored via video link using NHS software. Qualitative interviews will now be conducted via telephone for both patients and physiotherapists.

Follow-up data collection via telephone, and postal questionnaire data collection options have been added to minimise the need for participant hospital attendance. The study team will attempt to contact these participants on up to two occasions to remind them to complete the questionnaires. If these methods fail, we will categorise the participant as a 'non-responder' for that time point only. Qualitative data will now be collected using telephone interviews for all groups.

We plan to recruit an additional three participants to replace the participants recruited pre-COVID who were unable to adhere to the intervention due to the emergency changes in service provision.

## Patient and public involvement

Patient involvement began during protocol and intervention development and continues throughout the trial. A patient-member will attend all TSC meetings. The same patient member is a coinvestigator, providing insights into the trial conduct, particularly on data collection processes, and will help interpret the findings to inform on the implications of the research during the trial's dissemination phase.

## ETHICS AND DISSEMINATION

Ethical approval was gained from the South Central (Oxford B) Research Ethics Committee. A TSC was appointed to independently review the data on safety, protocol adherence and recruitment to the trial. Direct access will be granted to authorised representatives from the sponsor and host institution for monitoring and/or audit of the trial to ensure compliance with regulations. Anonymised data will be shared outside the research team when required. Researchers outside the trial team may formally request for a specific data set as per the Data Management Plan. All requests will need to be approved by the TMG.

Reporting of the trial will be consistent with the Consolidated Standards of Reporting Trials 2010 Statement and its various extensions (pilot and feasibility trials, patient reported outcomes and non-pharmacological interventions)[44] and Template for Intervention Description and Replication guidelines.[45] A summary of the results and trial materials will be made available via the trial website on completion of the trial. We will submit the final report to a peer-reviewed academic journal.

**Author affiliations**
[1]Nuffield Department of Orthopaedics, Rheumatology and Musculoskeletal Sciences (NDORMS), University of Oxford, Oxford, UK
[2]School of Health Sciences, University of East Anglia, Norwich, UK
[3]Oxford Clinical Trials Research Unit, Centre for Statistics in Medicine, Nuffield Department of Orthopaedics, Rheumatology and Musculoskeletal Sciences (NDORMS), University of Oxford, Oxford, UK
[4]College of Medicine and Health, University of Exeter, Exeter, UK
[5]Nuffield Department of Surgical Sciences, University of Oxford, Oxford, UK

**Collaborators** GRRAND-F Collaborators: Mrs Angela Garrett (Senior Trial Manager), Ms Alana Morris (Trial Manager), Mr Ray Derkacz (Patient and Public Member), Sites: Norfolk and Norwich University Hospitals NHS Foundation Trust (Principal Investigator: Mr Richard Sisson), Oxford University Hospital NHS Foundation Trust (Principal Investigator: Mr Stuart Winter), Poole Hospital NHS Foundation Trust (Principal Investigator: Ms Emma King) and University Hospitals Birmingham NHS Foundation Trust (Principal Investigator: Professor Hisham Mehanna).Trial Steering Committee: Dr Matthew Maddocks (Kings College London), Professor Vinidh Paleri (The Royal Marsden Hospital NHS Foundation Trust, London).This study will be conducted as part of the portfolio of trials in the registered UKCRC Oxford Clinical Trials Research Unit (OCTRU) at the University of Oxford. It will follow their Standard Operating Procedures ensuring compliance with the principles of Good Clinical Practice and the Declaration of Helsinki and any applicable regulatory requirements.

**Contributors** SCW, TOS, SL and SD researched the topic and devised the study. SCW, VG, TOS, SL, SD, MC-J and BF provided the first draft of the manuscript. SD provided statistical oversight. SCW, VG, TOS, SL, SD, MC-J and BF contributed equally to manuscript preparation. SCW acts a guarantor. All contributors approved the final version of the manuscript.

**Funding** This study is funded by the National Institute for Health Research (NIHR) Research for Patient Benefit grant (PB-PG-1217-20031). The views expressed are those of the authors and not necessarily those of the NIHR or the Department of Health and Social Care. Trial Sponsor: Oxford University Hospitals NHS Foundation Trust (OUH Research & Development, Joint Research Office, 2nd Floor, OUH Cowley, Unipart Buisness Centre, Garsington Road, Oxford, OX4 2PG. Email: OUH.Sponsorship@oxnet.nhs.uk.

**Competing interests** None declared.

**Patient consent for publication** Not required.

**Provenance and peer review** Not commissioned; externally peer reviewed.

of the author(s) and are not endorsed by BMJ. BMJ disclaims all liability and responsibility arising from any reliance placed on the content. Where the content includes any translated material, BMJ does not warrant the accuracy and reliability of the translations (including but not limited to local regulations, clinical guidelines, terminology, drug names and drug dosages), and is not responsible for any error and/or omissions arising from translation and adaptation or otherwise.

**ORCID iDs**
Victoria Gallyer http://orcid.org/0000-0002-2779-9364
Toby O Smith http://orcid.org/0000-0003-1673-2954

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
