## [Reviewer comments · BMJ Open]

ARTICLE DETAILS

TITLE (PROVISIONAL)	Getting Recovery Right After Neck Dissection (GRRAND-F): mixed-methods Feasibility study to design a pragmatic randomised controlled trial Protocol
AUTHORS	Gallyer, Victoria; Smith, Toby O.; Fordham, Beth; Dutton, Susan; Chester-Jones, Mae; Lamb, Sarah; Winter, Stuart

VERSION 1 – REVIEW

REVIEWER	Elise Gane The University of Queensland, and Princess Alexandra Hospital, Australia.
REVIEW RETURNED	05-Dec-2020

GENERAL COMMENTS	Getting recovery right after neck dissection (GRRAND-F): mixed-methods feasibility study to design a pragmatic randomised controlled trial protocol Thank you for the opportunity to review this protocol for a randomised controlled trial of rehabilitation after neck dissection surgery. There are many elements of this study methodology that are to be commended, particularly in the steps taken to train staff, avoid intervention contamination, measure co-intervention, etc. However, the use of a generalised rehab program makes it unclear how the researchers will identify which components are effective, and what needs to be changed for the larger scale RCT. Recruitment and treatment in the study is ongoing, so I hope the researchers find the comments below useful in reflecting on the results in the present study, and planning for the larger RCT in the future. Abstract – it's not clear if the intervention is rehab of the shoulder, neck, or in general. The Introduction presents this work as UK-centric, mentioning NICE guidelines. The guidelines call for prospective RCTs, and the Cochrane review does too. However the Cochrane review was published in 2012 and since then, the work of Aoife McGarvey from Australia has been published, which includes a prospective RCT of shoulder rehabilitation. Furthermore, the work of Margaret McNeely from Canada is missing – Prof McNeely has published the best RCT to date on rehabilitation of the shoulder after ND for HNC. These are critical papers in this field, and should be mentioned alongside this NICE guideline as background to this study. McGarvey AC, Hoffman GR, Osmotherly PG, Chiarelli PE. Maximizing shoulder function after accessory nerve injury and neck dissection surgery: A multicenter randomized controlled trial.
---

Head Neck. 2015 Jul;37(7):1022-31. doi: 10.1002/hed.23712. Epub 2014 Jul 11. PMID: 25042422.
McNeely, M.L., Parliament, M.B., Seikaly, H., Jha, N., Magee, D.J., Haykowsky, M.J. and Courneya, K.S. (2008), Effect of exercise on upper extremity pain and dysfunction in head and neck cancer survivors. *Cancer*, 113: 214-222.
<https://doi.org/10.1002/cncr.23536>

Do the authors know how many sacrificed spinal accessory nerves they will have? It's not very common anymore.

Exclusion criteria: Researchers have stated that they will exclude those with a pre-existing neurological condition that affects the shoulder, which is a good idea. What about pre-existing musculoskeletal conditions like history of recurrent shoulder dislocations or rotator cuff surgery? These are likely to affect baseline function and amount of functional shoulder recovery available post surgery. Researchers might consider whether it is feasible to collect this information from the participants they have already recruited.

Exclusion criteria: Are patients who undergo radiation therapy eligible? How is the intervention being modified to account for pain and fatigue associated with radiation therapy? Perhaps post-op radiation therapy could be considered as a covariate in the statistical analysis. My research has identified radiation therapy as a risk factor for poor shoulder dysfunction – beyond what occurs as a result of the ND itself. (Gane E et al (2017). Neck and upper limb dysfunction in patients following neck dissection: Looking beyond the shoulder. *Otolaryngology - Head and Neck Surgery*, 157 (4), 631-640. doi: 10.1177/0194599817721164)

Intervention: The aim of the study listed in the introduction is to “evaluate whether it is feasible to run a RCT to assess the effectiveness and cost-effectiveness of a rehabilitation intervention in improving pain, function and health-related quality of life following ND after HNC”. The aim does not provide either a thorough idea of the nature of the intervention, or details of the outcomes – pain where? What kind of function? General or ND specific HRQOL?

Having a multi-modal rehabilitation program is certainly a reflection of what actually occurs within clinical practice, but comes at the cost of knowing what part of the multi-modal intervention is the most effective at changing a specific outcome. Conducting an evaluation of a model of care rather than a rehabilitation intervention appears to be better suited to what the researchers are doing.

The nature of the resistance training in the intervention may not be effective in changing shoulder function, based on previous evidence. This study proposes to use resistance training with resistance bands progressing to weight bearing exercises for the shoulder, and isometric resistance for the neck and TMJ. By 6 months post-op, the shoulder is likely to be recovered enough for the patient to use free weights and weight machines. Without the progression to this level of resistance, chances in outcomes such as upper limb function may not be as pronounced as they otherwise would. For example, McGarvey's 2015 RCT published in *Head & Neck* recruited people within 8 weeks of ND surgery, and

	delivered 12 weeks of weekly F2F therapy plus 2 HEP sessions per week. Progressive scapular exercises were prescribed against the resistance of hand weights. If not hand weight, then resistance bands. Self-reported function improved over time in both the control and intervention groups, but did not differ between groups at any stage (SPADI). Active shoulder abduction ROM was higher in the intervention group at 3 months, but not at 6 or 12 months. Compare this to McNeely's study, which did use machine weights, and did achieve changes in shoulder function immediately post intervention between intervention and control groups. Caveat: patients began their rehab from between 2 months post op to 180 months post op. Outcomes: Shoulder and neck ROM is being measured (active or passive? Goniometer or inclinometer?), and resistance training prescribed, but why isn't shoulder and neck muscle strength being measured? How will the researchers know if the exercise was intense enough to induce physiological change in the muscle? Only self-reported shoulder function is being measured, however some of the intervention also targets the neck. Did the authors consider using the Neck Disability Index to measure self-reported pain-related neck function? Healthcare utilisation data: Good to see this data being collected – will provide useful information. Qualitative study: how will gender be addressed in purposive sampling, given HNC is more common in men than women? Or tumour sub-site?
--	--

REVIEWER	Joseph Dort University of Calgary, Canada
REVIEW RETURNED	16-Dec-2020

GENERAL COMMENTS	The manuscript is a proposal for a feasibility study of a randomized controlled trial in patients undergoing neck dissection for head and neck cancer. Participants will be randomized to either usual care or an enhanced / individualised rehabilitation protocol (GRRAND-F). Mixed methods are used in that both physical outcomes and quality of life will be measured. Feasibility will also be measured and will inform whether or not a larger RCT can / should be conducted. Reviewer Comments:  1) The protocol is clear and well-written 2) The topic is important and worthy of investigation 3) I am confused about the study cohort. The authors are looking at neck dissection and the index procedure but in reality patients in this study will have far more than neck dissection performed. In many cases there will be large resections of primary head and neck tumours (oral cavity, larynx, pharynx, etc) and there will also be reconstruction performed. Therefore it's not quite as simple as "neck dissection". The authors don't make it clear how the heterogeneity of surgical intervention will be managed nor do they indicate how this will affect sample size and analysis factors such as stratification and analysis of confounder / modifiers. 4) Many of the patients in this study will have postoperative adjuvant therapies (radiation +/- chemotherapy). The impact of these therapies on study outcomes are not discussed or accounted for.
---

	5) Overall I believe this is a worthy study but the issues raised must be addressed.
--	--

VERSION 1 – AUTHOR RESPONSE

Reviewer 1

Comment: Thank you for the opportunity to review this protocol for a randomised controlled trial of rehabilitation after neck dissection surgery. There are many elements of this study methodology that are to be commended, particularly in the steps taken to train staff, avoid intervention contamination, measure co-intervention, etc. However, the use of a generalised rehab program makes it unclear how the researchers will identify which components are effective, and what needs to be changed for the larger scale RCT. Recruitment and treatment in the study is ongoing, so I hope the researchers find the comments below useful in reflecting on the results in the present study, and planning for the larger RCT in the future.

Response: Thank you for your review of our paper. We have answered the issue regarding the generalised rehabilitation intervention in the responses below.

Comment: Abstract – it's not clear if the intervention is rehab of the shoulder, neck, or in general.

Response: We have now specified that this is a neck and upper limb exercise programme with education and advice (**Abstract, Methods and Analysis, Line 9-10**).

Comment: The Introduction presents this work as UK-centric, mentioning NICE guidelines. The guidelines call for prospective RCTs, and the Cochrane review does too. However the Cochrane review was published in 2012 and since then, the work of Aoife McGarvey from Australia has been published, which includes a prospective RCT of shoulder rehabilitation. Furthermore, the work of Margaret McNeely from Canada is missing – Prof McNeely has published the best RCT to date on rehabilitation of the shoulder after ND for HNC. These are critical papers in this field, and should be mentioned alongside this NICE guideline as background to this study.

McGarvey AC, Hoffman GR, Osmotherly PG, Chiarelli PE. Maximizing shoulder function after accessory nerve injury and neck dissection surgery: A multicenter randomized controlled trial. *Head Neck*. 2015 Jul;37(7):1022-31. doi: 10.1002/hed.23712. Epub 2014 Jul 11. PMID: 25042422.

McNeely, M.L., Parliament, M.B., Seikaly, H., Jha, N., Magee, D.J., Haykowsky, M.J. and Courneya, K.S. (2008), Effect of exercise on upper extremity pain and dysfunction in head and neck cancer survivors. *Cancer*, 113: 214-222. <https://doi.org/10.1002/cncr.23536>

Response: Thank you for highlighting these paper to be included in the protocol paper. These have been added into the Introduction as recommended to provide context to this study (**Introduction, Paragraph 5, Lines 9-11**).

Comment: Do the authors know how many sacrificed spinal accessory nerves they will have? It's not very common anymore.

Response: Sacrificing the accessory nerve is uncommon in current practice. It is not possible to say in a prospective study how many will be included. There is no national database for the incidence of accessory nerve sacrifice as part of a neck dissection. With the Oxford Head and Neck unit the incidence of sacrificing the accessory nerve is less than 2%. As a multi-centre trial practice will vary. This study will provide an estimate of the incidence. To clarify this, we have specifically acknowledged this in the revised data collection schedule (**Table 2**).

Comment: Exclusion criteria: Researchers have stated that they will exclude those with a pre-existing neurological condition that affects the shoulder, which is a good idea. What about pre-existing musculoskeletal conditions like history of recurrent shoulder dislocations or rotator cuff surgery? These are likely to affect baseline function and amount of functional shoulder recovery available post surgery. Researchers might consider whether it is feasible to collect this information from the participants they have already recruited.

Response: Participants who have a pre-existing musculoskeletal condition are eligible to be recruited. One aspect of the feasibility study is to explore whether the intervention is acceptable to these patients. We also want to assess the frequency to which these patients are recruited into this trial. This is important given the average age of a new patient who undergoes surgery for head and neck cancer in approximately 70 to 74 years of age. We are collecting pre-existing shoulder and neck musculoskeletal disorders as part of the demographic/profile characteristics. When used in addition to the baseline SPADI, we should have a clear picture on the potential impact this may have both in recruitment but also intervention acceptability/tolerance. We will also explore this issue in our qualitative interviews with patients and healthcare professionals as we agree that this is an important consideration. This has been highlighted in the qualitative topic guide which is a supplementary file in this resubmission (**Supplementary File 3**).

Comment: Exclusion criteria: Are patients who undergo radiation therapy eligible? How is the intervention being modified to account for pain and fatigue associated with radiation therapy? Perhaps post-op radiation therapy could be considered as a covariate in the statistical analysis. My research has identified radiation therapy as a risk factor for poor shoulder dysfunction – beyond what occurs as a result of the ND itself. (Gane E et al (2017). Neck and upper limb dysfunction in patients following neck dissection: Looking beyond the shoulder. *Otolaryngology - Head and Neck Surgery*, 157 (4), 631-640. doi: 10.1177/0194599817721164)

Response: This is a very good point. Respecting the individualised nature of the intervention, the decision on timing of treatments such as radiotherapy will be determined by the clinic-pathological results (lead by the clinical/non-research team). However the timing of the post-treatment rehabilitation in this trial, is based on the patient-physiotherapist shared decision making. This has been detailed in the revised paper to address this concern (**Methods and Analysis, Experimental Intervention, Delivery, Paragraph 2, Lines 1-8**).

Given the objective of the study and sample size, we will record the number of people receiving radiotherapy/chemotherapy during the study period, and the timing of these to intervention but do not plan to use these as covariates of the analysis of this feasibility study but will consider whether there is added value in adding this as a stratification or a covariate in the definitive study which will follow if this study is shown to be feasible.

Comment: Intervention: The aim of the study listed in the introduction is to “evaluate whether it is feasible to run a RCT to assess the effectiveness and cost-effectiveness of a rehabilitation intervention in improving pain, function and health-related quality of life following ND after HNC”. The aim does not provide either a thorough idea of the nature of the intervention, or details of the outcomes – pain where? What kind of function? General or ND specific HRQOL?

Response: Thank you for this comment. This relates to the text in the Introduction (**Paragraph 6, Lines 1-3**). We have reflected on this, and how it is placed in the whole paper. A fuller outline which answers this point is presented in the following paragraph and subsequent methods section. We therefore do not wish to repeat the same text throughout the paper. Nonetheless, we have made slight changes to ensure more detail is presented on the intervention but feel the reader can ascertain further detail on the nature of the intervention and outcomes shortly afterwards in the Methods and Analysis section (**Introduction, Paragraph 6, Lines 1-2**). We hope that this approach is acceptable to Reviewer 1 and the Editor.

Comment: Having a multi-modal rehabilitation program is certainly a reflection of what actually occurs within clinical practice, but comes at the cost of knowing what part of the multi-modal intervention is the most effective at changing a specific outcome. Conducting an evaluation of a model of care rather than a rehabilitation intervention appears to be better suited to what the researchers are doing.

Response: Thank you for this comment. The GRRAND intervention was developed to be theoretically delivered in the NHS (and similar healthcare models). The intention of this programme of research is to firstly explore, once demonstrated to be feasible, the clinical and cost-effectiveness of this intention in a pragmatic randomised controlled trial. We would then anticipate an evaluation of the specific components in the multi-modal intervention using mediation analysis. We feel it is not appropriate to explain this future proposed step, until the results of this feasibility study are known. Given this, we currently elect not to amend the text in relation to this point.

Comment: The nature of the resistance training in the intervention may not be effective in changing shoulder function, based on previous evidence. This study proposes to use resistance training with resistance bands progressing to weight bearing exercises for the shoulder, and isometric resistance for the neck and TMJ. By 6 months post-op, the shoulder is likely to be recovered enough for the patient to use free weights and weight machines. Without the progression to this level of resistance, chances in outcomes such as upper limb function may not be as pronounced as they otherwise would. For example, McGarvey’s 2015 RCT published in Head & Neck recruited people within 8 weeks of ND surgery, and delivered 12 weeks of weekly F2F therapy plus 2 HEP sessions per week. Progressive scapular exercises were prescribed against the resistance of hand weights. If not hand weight, then resistance bands. Self-reported function improved over time in both the control and intervention groups, but did not differ between groups at any stage (SPADI). Active shoulder abduction ROM was higher in the intervention group at 3 months, but not at 6 or 12 months. Compare this to McNeely’s study, which did use machine weights, and did achieve changes in shoulder function

immediately post intervention between intervention and control groups. Caveat: patients began their rehab from between 2 months post op to 180 months post op.

Response: Thank you for this study. We would hope to be able to test these suggestions in a definite trial. We will be unable to answer this question in this feasibility study since it is not our objective to do so, merely to assess the feasibility of the study design, and acceptability/fidelity of the intervention. Whilst we understand the reviewer's important points, we hope the multi-modal intervention will have a benefit on exercise prescription and health belief on recovery and symptom management which may offer differences in the recovery profiles reported in previous trials. Coupled with the difference in timing of interventions, we believe the GRRAND intervention may provide new and novel findings in contrast to McGarvey and other's previous work. Given that it is not the purpose of this feasibility study to explore intervention effectiveness, no amendment is required in the text to address this comment raised.

Comment: Outcomes: Shoulder and neck ROM is being measured (active or passive? Goniometer or inclinometer?), and resistance training prescribed, but why isn't shoulder and neck muscle strength being measured? How will the researchers know if the exercise was intense enough to induce physiological change in the muscle? Only self-reported shoulder function is being measured, however some of the intervention also targets the neck. Did the authors consider using the Neck Disability Index to measure self-reported pain-related neck function?

Response: Thank you for these questions. The decision regarding the outcome set used in this feasibility study was based on patient and public involvement view points on important domains of assessment, balanced against the overarching assessment of effectiveness which the proposed definitive trial would be. This was adopted in the absence of a published core outcome set in this population. Whilst we considered shoulder and neck muscle strength, a potentially important measure, given that the domain of function is comprehensively measured through the patient-reported outcome measures used, we elected not to assess these in this study. We also needed to balance the potential burden of outcome measures being provided to participants to complete and the risk that this could have on attrition and missing data. Given the number currently offered, it was elected not to include a measure of neck function, with the overall Health-Related Quality of Life with the EORTC questionnaire C30 core [25] and H&N43 (head and neck specific)[26,27] questionnaires covering (in part) this domain. Nonetheless, we acknowledge this point and will explore the selection of outcome measures as part of the qualitative study of both healthcare professionals and patients. This has been highlighted in the text (**Supplementary File 3**). Through this approach, if muscle strength and neck function are highlighted as key domains for examination, this finding will be reported and included in a future definitive trial.

Comment: Healthcare utilisation data: Good to see this data being collected – will provide useful information.

Response: Thank you for your supportive comment. No amendment required to the text.

Comment: Qualitative study: how will gender be addressed in purposive sampling, given HNC is more common in men than women? Or tumour sub-site?

Response: We have ensured that this is presented for the reader in the paper detailing that participants who have agreed to be contacted for the interview will be purposively sampled by the qualitative researcher to ensure the 15 interview participants are demographically representative of the full study sample. Targeted demographics include age, ethnicity, employment status, and extent of neck dissection. (**Methods and Analysis; Qualitative Investigation; Recruitment; Paragraph 3; Lines 1-4**).

We will purposively sample to ensure the interviewees are demographically representative of the full study sample. We estimate that the study sample will include more males than females because 69% of HNC cases in the UK are in males and 31% are in females. We aim to invite two males for every one female we interview. However, if we are restricted in the number of participants available for interview then we will interview as many as available. We will highlight the sex of the participants as part of our interpretation of our qualitative analysis. This has been presented in the revised paper (**Methods and Analysis; Qualitative Investigation; Recruitment; Paragraph 3; Lines 4-9**).

Reviewer 2

Comment: The manuscript is a proposal for a feasibility study of a randomized controlled trial in patients undergoing neck dissection for head and neck cancer. Participants will be randomized to either usual care or an enhanced / individualised rehabilitation protocol (GRRAND-F). Mixed methods are used in that both physical outcomes and quality of life will be measured. Feasibility will also be measured and will inform whether or not a larger RCT can / should be conducted.

Response: Thank you for the accurate summary of the study.

Comment: 1) The protocol is clear and well-written

Response: Thank you for your kind words. No amendment required in response.

Comment: 2) The topic is important and worthy of investigation.

Response: Thank you for your kind words. No amendment required in response.

Comment: 3) I am confused about the study cohort. The authors are looking at neck dissection and the index procedure but in reality patients in this study will have far more than neck dissection performed. In many cases there will be large resections of primary head and neck tumours (oral cavity, larynx, pharynx, etc) and there will also be reconstruction performed. Therefore it's not quite as simple as "neck dissection". The authors don't make it clear how the heterogeneity of surgical

intervention will be managed nor do they indicate how this will affect sample size and analysis factors such as stratification and analysis of confounder / modifiers.

Response: The reviewers correctly identify the challenges of performing research in this area. There is an inherent heterogeneity to the study population and their treatment. The study has been designed to be inclusive to the Head and Neck population as a whole. The intervention has been designed so that it can be delivered in up to 6 sessions with the therapy addressing the needs of the patients. The title reflects that all patients will have had a neck dissection, but the intervention allows for a broader remit of rehabilitation. This is a feasibility study which will explore the delivery of the intervention as well as the feasibility of recruitment and retention. Subject to a successful result then the aim will be to seek funding for a larger definitive study that can assess outcome and cost effectiveness. This has been acknowledged in the revised text (**Methods and Analysis, Eligibility, Lines 2-8**).

Comment: 4) Many of the patients in this study will have postoperative adjuvant therapies (radiation +/- chemotherapy). The impact of these therapies on study outcomes are not discussed or accounted for.

Response: Thank you for highlighting this important point. We have acknowledged in the revised paper, the impact that adjuvant therapies will have on the timing and delivery of the intervention (**Methods and Analysis, Experimental Intervention, Delivery, Paragraph 2, Lines 1-8**). We will also be monitoring the frequency and variation of these adjuvant therapies on our feasibility study cohort as part of the data collection process (**Table 2**). This will also be explored in our qualitative study, as illustrated in the now included qualitative topic guide (**Supplementary File 3**). Through this, we will be better informed on the impact of these on our participants and study design to inform the development of a definitive trial.

Comment: 5) Overall I believe this is a worthy study but the issues raised must be addressed.

Response: Thank you for your kind words. No amendment required in response.

VERSION 2 – REVIEW

REVIEWER	Gane, Elise University of Queensland, School of Health and Rehabilitation Sciences
REVIEW RETURNED	01-Apr-2021
GENERAL COMMENTS	Thank you for addressing the reviewer queries comprehensively. No further changes are recommended.